# CORESET SPECTRAL CLUSTERING

Ben Jourdan[1], Gregory Schwartzman[2], Peter Macgregor[3], and He Sun[1]

[1]University of Edinburgh, UK
[2]Japan Advanced Institute of Science and Technology (JAIST), Japan
[3]University of St Andrews, UK
ben.jourdan@ed.ac.uk, greg@jaist.ac.jp, prm4@st-andrews.ac.uk, h.sun@ed.ac.uk

## ABSTRACT

Coresets have become an invaluable tool for solving $k$-means and kernel $k$-means clustering problems on large datasets with small numbers of clusters. On the other hand, spectral clustering works well on sparse graphs and has recently been extended to scale efficiently to large numbers of clusters. We exploit the connection between kernel $k$-means and the normalised cut problem to combine the benefits of both. Our main result is a coreset spectral clustering algorithm for graphs that clusters a coreset graph to infer a good labelling of the original graph. We prove that an $\alpha$-approximation for the normalised cut problem on the coreset graph is an $O(\alpha)$-approximation on the original. We also improve the running time of the state-of-the-art coreset algorithm for kernel $k$-means on sparse kernels, from $\tilde{O}(nk)$ to $\tilde{O}(n \cdot \min\{k, d_{avg}\})$, where $d_{avg}$ is the average number of non-zero entries in each row of the $n \times n$ kernel matrix. Our experiments confirm our coreset algorithm is asymptotically faster on large real-world graphs with many clusters, and show that our clustering algorithm overcomes the main challenge faced by coreset kernel $k$-means on sparse kernels which is getting stuck in local optima.

## 1 INTRODUCTION

Kernel $k$-means and spectral clustering are two popular algorithms which are capable of learning non-linear decision boundaries between clusters. For this reason, they have been applied to many practical problems in machine learning, including in the fields of medical research and network science (Gönen & Margolin, 2014; Kuo et al., 2014; White & Smyth, 2005).

Given a data set $X$, both kernel $k$-means and spectral clustering make use of a kernel similarity function $K : X \times X \to \mathbb{R}_{\geq 0}$, which is often represented as a matrix of size $n \times n$, where $n = |X|$. The spectral clustering algorithm considers the kernel matrix to be the adjacency matrix of a *similarity graph* and clusters the nodes of the graph in order to minimise the *normalised cut* objective function, which we define in Section 3 (Von Luxburg, 2007). Kernel $k$-means exploits the fact that the kernel function implicitly defines an embedding of the data points into a Hilbert space, represented by $\phi : X \to \mathcal{H}$, such that for all $x, y \in X$,

$$\langle \phi(x), \phi(y) \rangle = K(x, y). \tag{1}$$

The kernel $k$-means problem is to minimise the $k$-means objective in this Hilbert space and is usually solved using a generalisation of Lloyds algorithm (Dhillon et al., 2004), using the kernel $K$ to compute inner products. Remarkably, the normalised cut and kernel $k$-means objectives are equivalent up to a constant factor (Dhillon et al., 2004).

Despite this equivalence, spectral clustering and kernel $k$-means have been largely studied separately, and new techniques have been developed with one or the other in mind. For spectral clustering, one of the most promising techniques is to construct a sparse similarity graph in order to achieve a speedup. This can be achieved through constructing an approximate $k$-nearest neighbour graph using fast approximate nearest neighbour algorithms (Alshammari et al., 2021; Harwood & Drummond, 2016; Malkov & Yashunin, 2018), or by constructing a clustering-preserving sparsifier of the complete kernel similarity graph (Macgregor & Sun, 2023; Sun & Zanetti, 2019). By operating on a sparse similarity graph, the time and memory cost of spectral clustering is reduced from $\Omega(n^2)$ to $O(n \log(n))$.

To speed up kernel $k$-means, Jiang et al. (2024) applied a recent coreset result for Euclidean spaces (Braverman et al., 2021) to kernel spaces. Given a dataset $X$ and kernel function $K$, they showed that an $\varepsilon$-coreset (Definition 4) of size $\tilde{O}(k^2\varepsilon^{-4})$ can be constructed in time $\tilde{O}(nk)$[1]. The cost of running an iteration of Lloyd's algorithm on the coreset is then independent of $n$, for small $k$.

In this paper, we show that the benefits of the techniques developed for both spectral clustering and kernel $k$-means can be combined in order to create a faster and more accurate clustering algorithm. In particular, we show that, by exploiting the sparsity of the kernels usually considered for spectral clustering, it is possible to design a faster algorithm for constructing a coreset. Then, we apply spectral clustering directly to the coreset graph with only a small number of nodes and use the result to cluster the original graph. Empirically, we find that on sparse graphs our coreset spectral clustering algorithm has a significantly faster running time than the classical spectral clustering algorithm, and achieves a higher accuracy than coreset kernel $k$-means.

## 1.1 OUR RESULTS

**Faster $k$-means++ and coreset construction.**     In Section 4, we exploit the fact that inputs to kernel $k$-means are often sparse to devise a faster algorithm for $k$-means++ initialisation in kernel space. Specifically, when the input kernel matrix is sparse, we improve the running time from $\tilde{O}(nk)$ to $\tilde{O}(n \cdot \min\{k, d_{avg}\})$ where $d_{avg}$ is the average number of non-zero entries in each row of the kernel matrix. Speeding up $k$-means++ has received a lot of attention for Euclidean spaces (Bachem et al., 2016; Cohen-Addad et al., 2020). This is the first result to provide speed up in sparse kernel spaces. Combining this with Jiang et al. (2024), we present the first coreset construction for kernel spaces that makes use of kernel sparsity, and reduce the coreset construction time from $\tilde{O}(nk)$ to $\tilde{O}(n \cdot \min\{k, d_{avg}\})$ while maintaining the same theoretical guarantees. For large graphs with many clusters, it is critical to break the linear dependence on the number of clusters for practical use.

**Coreset spectral clustering.**     In Section 5, we introduce the coreset spectral clustering algorithm that explicitly combines spectral clustering with coresets by exploiting the equivalence between the normalised cut and weighted kernel $k$-means problems. Previous work has used this equivalence to convert spectral clustering problems to kernel $k$-means problems and then solve them using coreset kernel $k$-means (Jiang et al., 2024). We propose a new method which solves the coreset problem directly with spectral clustering and then transfers the solution back to the original graph. This sidesteps some of the less desirable properties that running kernel $k$-means would entail, such as its susceptibility to local minima when using indefinite kernels (Dhillon et al., 2007). In Theorem 1, we prove that an $\alpha$-approximation of the normalised cut problem on the coreset graph gives an $O(\alpha)$-approximation of the normalised cut problem on the original graph.

**Experiments.**     In Section 6, we preform three experiments to test our coreset construction and coreset spectral clustering algorithms. The first confirms the asymptotic improvement in running time of our coreset construction algorithm for kernel $k$-means over the previous method of Jiang et al. (2024) on large real-world graphs with up to 65 million nodes and thousands of clusters. The second experiment compares our coreset spectral clustering algorithm against coreset kernel $k$-means (Jiang et al., 2024) and the sklearn (Pedregosa et al., 2011) implementation of spectral clustering on several real-world graph datasets. This shows that for a small number of clusters, the coreset methods are much faster than spectral clustering and our coreset spectral clustering algorithm finds significantly better solutions than coreset kernel $k$-means. The third experiment compares our coreset spectral clustering algorithm against coreset kernel $k$-means on a synthetic graph dataset where we vary the number of clusters to be linear in the number of nodes. Using the spectral clustering method proposed by Macgregor (2023) to cluster the coreset graphs, this experiment shows that our method can scale to hundreds of clusters while coreset kernel $k$-means is rendered ineffective by local optima after only tens of clusters.

## 2 RELATED WORK

The techniques for speeding up kernel $k$-means and spectral clustering can be broadly categorised into the methods that sparsify the relations between input data and the methods based on coresets.

---

[1]We use $\tilde{O}(\cdot)$ to suppress polylogarithmic factors.

For kernel $k$-means (Dhillon et al., 2004), low rank approximations of the kernel matrix have been proposed (Musco & Musco, 2017; Wang et al., 2019) as well as low dimensional approximations of $\phi$ (Chitta et al., 2012). On the other hand, coresets for Euclidean spaces (Braverman et al., 2021; Har-Peled & Mazumdar, 2004; Huang & Vishnoi, 2020) have recently been extended to kernel $k$-means (Jiang et al., 2024). Coreset methods sample a weighted subset of the input so that the objective of every feasible solution is preserved.

For spectral clustering (Von Luxburg, 2007), spectral sparsifiers (Spielman & Teng, 2011) and more recently cluster preserving sparsifiers (Sun & Zanetti, 2019) can sparsify dense graphs while retaining cluster structure. While effective in theory, spectral sparsifiers require complicated Laplacian solvers to run efficiently, making them difficult to implement in practice. Cluster preserving sparsifiers are practical to implement but their performance is sensitive to the choice of hyperparameters. Peng et al. (2017) proposed a nearly-linear time algorithm for clustering an arbitrary number of clusters which is similar in spirit to a coreset approach. They sample $\Theta(k \log(k))$ nodes leveraging the property that in the spectral embedding nodes in the same cluster are nearby and have approximately the same norm. From this, they efficiently extract a set of $k$ points from which the rest of the data can be labelled. However, this approach is impractical as they make use of Laplacian solvers to approximate heat kernel distances.

As well as sparsification and coresets, speedup can also be achieved via improvements to the optimisation algorithms themselves. The triangle inequality can be used to reduce the number of inner products calculated by the kernel $k$-means algorithm (Dhillon et al., 2004) and recently a mini-batch algorithm has been proposed (Jourdan & Schwartzman, 2024). Macgregor (2023) developed a simple spectral clustering algorithm that forgoes the expensive computation of $k$ eigenvectors in place of $O(\log(k))$ independent calls to the power method. We refer to this method as fast spectral clustering and will use it in our experiments to cluster coreset graphs.

## 3 PRELIMINARIES

Let $X$ be a set of $n$ objects, and $K : X \times X \to \mathbb{R}_{\geq 0}$ be a function measuring the pairwise similarity of data points in $X$. Let $\phi : X \to \mathcal{H}$ be the function implicitly defined by $K$ that maps data points in $X$ to the unique Hilbert space such that $\langle \phi(x), \phi(y) \rangle = K(x, y)$ for all $x, y \in X$. The function $K$ is usually represented as a matrix: if $\Phi = [\phi(x_1), \ldots, \phi(x_n)]$, then $K = \Phi^T \Phi \in \mathbb{R}^{n \times n}$ with $K_{ij} = \langle \phi(x_i), \phi(x_j) \rangle$. Let $\Delta(\phi(x), \phi(y)) \triangleq \|\phi(x) - \phi(y)\|^2$ denote the squared distance in feature space for all $x, y \in X$, and $\Delta(\phi(x), C) = \min_{c \in C} \Delta(\phi(x), \phi(c))$ denote the smallest squared distance from $x \in X$ to a set $C \subseteq X$. We refer to the diagonal elements of $K$ as self similarities and say $x$ is a neighbour of (or incident to) $y$, written $x \sim y$, iff $\langle \phi(x), \phi(y) \rangle \neq 0$. If $x \in X$ and $S \subset X$, we say $x$ is incident to $S$ iff $x$ is incident to at least one element of $S$.

Given a graph $G = (V, E)$, the conductance of a set $S$ of vertices is defined as $\Phi_G(S) \triangleq |E(S, V \setminus S)| / \text{vol}(S)$ where $|E(S, V \setminus S)|$ is the number of edges crossing the cut between $S$ and $V \setminus S$ and $\text{vol}(S)$ is the total weight of edges incident to $S$. We define a $k$-partition of $X$ to be a collection of sets $\Pi = \{\pi_j\}_{j=1}^k$ such that each element of $X$ appears in exactly one member of $\Pi$.

We make extensive use of the following concepts in our analysis.

**Definition 1** (centroids)**.** *Given a $k$-partition $\Pi = \{\pi_j\}_{j=1}^k$ of a set $X$, a map $\phi : X \to \mathcal{H}$ for some Hilbert space $\mathcal{H}$, and a weight function $X \to \mathbb{R}_+$, define the set of centroids of $\Pi$ as $c_w^\phi(\Pi) \triangleq \{c_w^\phi(\pi_j)\}_{j=1}^k$ where $c_w^\phi(\pi_j) = \left( \sum_{x \in \pi_j} w(x)\phi(x) \right) / \left( \sum_{x \in \pi_j} w(x) \right)$.*

**Definition 2** (kernel $k$-means objective)**.** *Given a weighted dataset $X$ with weights $w : X \to \mathbb{R}_+$ and feature map $\phi : X \to \mathcal{H}$ satisfying equation 1 for some kernel function $K$, the weighted kernel $k$-means objective with respect to an arbitrary set of points $C \subseteq \mathcal{H}$ is*

$$\text{COST}_w^\phi(X, C) = \sum_{x \in X} w(x)\Delta(\phi(x), C). \tag{2}$$

*The kernel $k$-means objective with respect to an arbitrary $k$-partiton $\Pi = \{\pi_j\}_{j=1}^k$ of $X$ is*

$$\text{COST}_w^\phi(X, c_w^\phi(\Pi)) = \sum_{j=1}^k \sum_{x \in \pi_j} w(x)\Delta(\phi(x), c_w^\phi(\pi_j)),$$

**Definition 3** (Normalised cut objective). *Given a graph $G = (V, E)$, the normalised cut problem is to minimise the average conductance over all $k$-partitions of the vertices:* $\min_{\Pi=\{\pi_1,\dots,\pi_k\}} \text{NC}(G, \Pi)$,

*where* $\text{NC}(G, \Pi) = \frac{1}{k} \sum_{j=1}^{k} \Phi_G(\pi_j)$.

**Definition 4** ($\varepsilon$-coresets). *For $0 < \varepsilon < 1$, an $\varepsilon$-corset for kernel $k$-means on a weighted dataset $X$ with weights $w : X \to \mathbb{R}_+$ is a reweighted subset $S \subseteq X$ such that for the Hilbert space $\mathcal{H}$ and map $\phi : X \to \mathcal{H}$ satisfying equation 1, we have*

$$\text{COST}_{w'}^{\phi}(S, C) \in (1 \pm \varepsilon) \cdot \text{COST}_w^{\phi}(X, C), \quad \forall C \subset \mathcal{H} \text{ with } |C| = k,$$

*where* $w' : S \to \mathbb{R}_+$ *gives the weight for each element in the coreset.*

## 4   FAST CORESET CONSTRUCTION

Given a sparse kernel matrix $K$ with $O(m)$ nonzero entries and weight function $w$, we give an algorithm to construct an $\varepsilon$-coreset in $\tilde{O}(m)$ time. This improves on the algorithm given by Jiang et al. (2024) which has running time $\tilde{O}(nk)$. The running time of Jiang et al. (2024) is dominated by the running time of $D^2$-sampling, Algorithm 1, which is just the $k$-means++ initialisation algorithm (Arthur & Vassilvitskii, 2007) in kernel space. For completeness, we provide the full description of the $\varepsilon$-coreset algorithm of Jiang et al. (2024) in Algorithm 4 in Appendix C.1.

In Algorithm 2, we use the fact the kernel matrix is sparse to design an efficient data structure, built around a sampling tree (Wong & Easton, 1980), to perform $D^2$-sampling in nearly-linear time in $n$ and independent of $k$. To avoid unnecessary updates to the sampling tree, our algorithm has to sample one additional data point prior to performing $D^2$-sampling. Specifically, before uniformly sampling the first point, we will select the point in $C$ with the smallest self affinity. Let $x^* \triangleq \arg\min_{x \in X} \langle \phi(x), \phi(x) \rangle$ and $c^* \triangleq \langle \phi(x^*), \phi(x^*) \rangle$. We show that adding this extra point does not affect the approximation guarantee, and consequently we get a faster $\varepsilon$-coreset algorithm for sparse kernels using the same analysis as Jiang et al. (2024).

---

**Algorithm 1** $D^2$-Sampling($X$) (Jiang et al., 2024)

---

1: **Input:** $X$
2: Draw $x \in X$ uniformly at random, and initialise $C \leftarrow \{\varphi(x)\}$
3: **for** $i = 1, \dots, k-1$ **do**
4:     Draw one sample $x \in X$, using probabilities $w_X(x) \cdot \frac{\Delta(\varphi(x), C)}{\text{COST}_{w_X}^{\phi}(X,C)}$
5:     Let $C \leftarrow C \cup \{\varphi(x)\}$
6: **end for**
7: **return** $C$

---

### 4.1   OUR DATA STRUCTURE

The key observation is that as points are added to $C$, the distances from the data points to $C$ can't change for too many points due to the sparsity of the kernel and our choice of $x^*$. We formalise this intuition in what follows. Let $C \subset X$ be a set of points such that $x^*$ is in $C$, and consider the squared distance from an arbitrary point $x$ to $C$ in feature space. We see that

$$\begin{aligned}
\Delta(x, C) &= \min_{c \in C} \Big( \langle \phi(x), \phi(x) \rangle + \langle \phi(c), \phi(c) \rangle - 2\langle \phi(x), \phi(c) \rangle \Big) \\
&= \min_{c \in C} \begin{cases} \langle \phi(x), \phi(x) \rangle + \langle \phi(c), \phi(c) \rangle & x \nsim c \\ \langle \phi(x), \phi(x) \rangle + \langle \phi(c), \phi(c) \rangle - 2\langle \phi(x), \phi(c) \rangle & x \sim c \end{cases} \\
&= \begin{cases} \Delta(x, x^*) & x \nsim c \text{ for all } c \in C \\ \min_{c \in C} \Delta(x, c) & \text{otherwise,} \end{cases}
\end{aligned} \tag{3}$$

where the last transition makes use of the fact that $x^* \in C$ and

$$\Delta(x, x^*) = \langle \phi(x), \phi(x) \rangle + \langle \phi(x^*), \phi(x^*) \rangle \le \langle \phi(x), \phi(x) \rangle + \langle \phi(c), \phi(c) \rangle = \Delta(x, c)$$

for all $x \in X$ and $c \in C$ such that $x \nsim c$. From this we can see how adding a point to $C$ changes the distance between some $x \in X$ and $C$:

$$\Delta(x, C \cup \{y\}) = \begin{cases} \Delta(x, C) & x \nsim y \\ \min(\Delta(x, C), \Delta(x, y)) & x \sim y. \end{cases}$$

Assuming we know the values of $\Delta(x, C)$ for all $x \in X$, each time we add a point $y$ to $C$, for each neighbour $x$ of $y$, it suffices to check whether $\Delta(x, y)$ is less than $\Delta(x, C)$. Let $A \subset X$ be the set returned by Algorithm 2. Then the total number of checks is $\sum_{x \in A} \deg(x)$ where $\deg(x) = |\{y : x \sim y\}|$ is the number of nonzero entries in the row of $K$ corresponding to $x$. In the worst case, we have to check the entries in the rows of the $k + 1$ data points with the highest degree. Letting $d_{avg} = \frac{2|E|}{n}$ be the average degree, it holds that

$$d_{avg} = \frac{2|E|}{n} = \frac{1}{n} \sum_{x \in X} \deg(x) \geq \frac{1}{n} \sum_{x \in A} \deg(x),$$

and therefore $\sum_{x \in A} \deg(x) \leq n \cdot \deg_{avg}$ and the maximum number of checks we need to perform is $O(\min(d_{avg}, k) \cdot n)$.

**Sampling data points according to contribution.** We define the contribution of a data point $x$ with respect to $C$ as $f(x, C) \triangleq w_X(x) \cdot \Delta(\phi(x), C)$, and the contribution of a set of data points $S \subseteq X$ with respect to $C$ as $f(S, C) \triangleq \sum_{x \in S} f(x, C)$. Normalised by $\text{COST}_{w_X}^{\phi}(X, C)$, the contribution of each data point follows the $D^2$-sampling distribution and $\frac{f(S, C)}{f(X, C)}$ is the probability of sampling a point from a set $S \subseteq X$.

Suppose we have nested sets $S_1 \subset S_2 \subset S_3 \subset \cdots \subset S_m \subseteq X$. For any $x \in S_1$, we have that

$$\Pr[x \text{ is sampled}] = \frac{f(x, C)}{f(X, C)} = \frac{f(S_m, C)}{f(X, C)} \frac{f(S_{m-1}, C)}{f(S_m, C)} \cdots \frac{f(x, C)}{f(S_1, C)}. \tag{4}$$

Due to this decomposition, we can use a sampling tree $T$ to efficiently sample proportional to contributions and update contributions as points are added to $C$.

The root node of $T$ corresponds to the entire set $X$ and stores the total contribution $f(X, C)$. Sibling nodes represent a partition of the set their parent corresponds to and store their respective contributions. Leaf nodes store the data points they represent, their contributions, and weights.

To sample a data point proportional to contribution, we start at the root node of $T$ and recursively sample a child node with probability equal to the child's contribution divided by the parent's contribution until we reach a leaf, corresponding to the sampled data point. From equation 4, this is equivalent to sampling points porportional to contribution, and therefore the distribution required by $D^2$-sampling.

**Updating contributions efficiently.** We can update the contributions stored in $T$ efficiently each time a point $y$ is added to $C$. Suppose we find a neighbouring data point $x$ of $y$ such that $\Delta(x, y)$ is less than the stored value of $\Delta(x, C)$. Then we set the stored value of $\Delta(x, C)$ to be $\Delta(x, y)$ and subtract the difference in contribution from every internal node along the path from the leaf node to the root. If $T$ is balanced, then the cost to repair $T$ after each check is $O(\log(n))$. The full algorithm is given in Algorithm 2. Since $x^*$ can be found in time $O(n)$, $T$ is constructed in time $O(n)$, and the cost of sampling $k$ points and repairing $T$ is $O(\min(d_{avg}, k) \cdot n \log(n))$, the overall running time of Algorithm 2 is $\tilde{O}(\min(d_{avg}, k) \cdot n)$.

**Correctness.** We start with the following useful definition:

**Definition 5** (($\alpha, \beta$)-approximation for weighted kernel $k$-means)**.** *Let $OPT_k(X, \phi)$ denote the optimal objective of equation 2. We call a subset $S \subset \mathcal{H}$ an ($\alpha, \beta$)-approximate solution for kernel $k$-means if $|S| = \alpha k$ and $\text{COST}_w^{\phi}(X, S) \leq \beta OPT_k(X, \phi)$.*

$D^2$-sampling returns an $(O(1), O(\log(k)))$-approximate solution for $k$-means with high probability (Jiang et al., 2024); adding the extra point $x^*$ has a negligible effect. The proof follows that of Lemma 3.1 in Arthur & Vassilvitskii (2007) with the only difference being the equality of Lemma 3.1 becomes an inequality. We summarise the result in the following lemma:

**Lemma 4.1.** *Given a kernel matrix $K$ corresponding to dataset $X$, Algorithm 2 returns an $(O(1), O(\log k))$-approximation for kernel $k$-means with high probability and running time $\tilde{O}(\min(d_{avg}, k) \cdot n)$.*

By the analysis of Jiang et al. (2024), this immediately implies the following kernel coreset result:

**Lemma 4.2.** *Given a kernel matrix $K$ corresponding to dataset $X$, Algorithm 4, using Algorithm 2 for $D^2$-sampling, returns an $\varepsilon$-coreset with high probability with size $\tilde{O}(k^2\varepsilon^{-4})$ and running time $\tilde{O}(\min(d_{avg}, k) \cdot n)$.*

---

**Algorithm 2** FAST$D^2$-SAMPLING

---

1: **Input:** Dataset $X$ with $|X| = n$, positive definite matrix $K \in \mathbb{R}_{\geq 0}^{n \times n}$, $W \in \mathbb{R}_+^n$, $k \in \mathbb{N}$
2: $T \leftarrow$ CONSTRUCT$T(X, K, W)$             ▷ Algorithm 6
3: $x^* \leftarrow \arg\min_{x \in X} \langle \phi(x), \phi(x) \rangle$
4: draw $x \in X$ uniformly at random
5: $C \leftarrow \{\phi(x), \phi(x^*)\}$
6: REPAIR$(x^*, T)$                     ▷ Algorithm 7
7: **for** $i = 1$ to $k$ **do**
8:     $x \leftarrow$ SAMPLEPOINT$(T)$          ▷ Algorithm 8
9:     $C \leftarrow C \cup \{\phi(x)\}$
10:    REPAIR$(x, T)$
11: **end for**
12: **return** $C$

---

## 5   CORESET SPECTRAL CLUSTERING

In this section we present our Coreset Spectral Clustering (CSC) algorithm. An intuitive illustration is given in Figure 1. Given an input graph, we first extract the corresponding weighted kernel $k$-means problem via the equivalence to the normalised cut problem, and then construct an $\varepsilon$-coreset. Following this, again via the equivalence, we solve the corresponding normalised cut problem on the coreset graph to get the coreset partition. Finally, we label the rest of the data by considering kernel distances to the implied centers induced by the coreset graph partition. Our main contribution is summarised in Theorem 1, which states that coreset graphs preserve normalised cuts from the original graph.

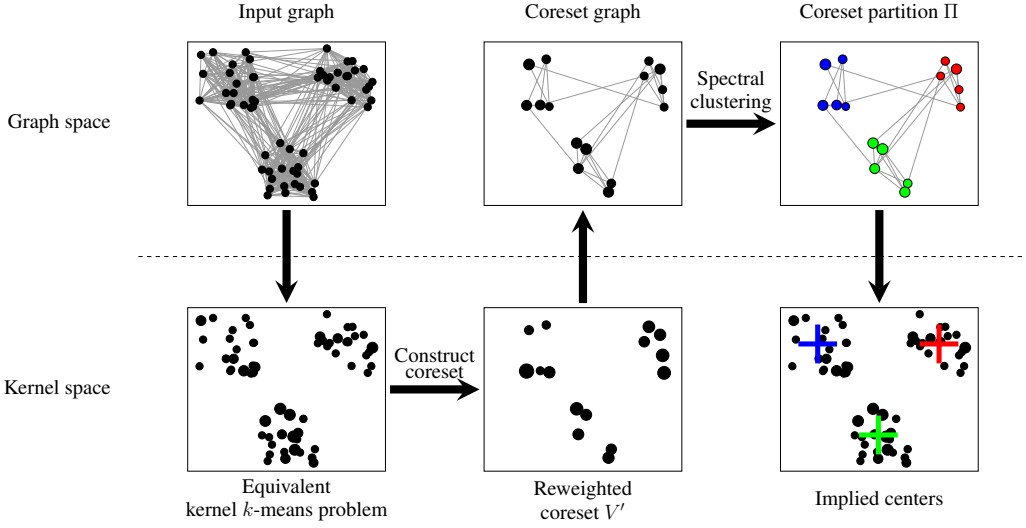

Figure 1: Sketch of the Coreset Spectral Clustering Algorithm.

**Equivalence between normalised cut and kernel $k$-means.** First recall that the normalised cut problem and the kernel $k$-means problem can both be written as the following trace optimisation problems up to a constant (Dhillon et al., 2004):

| **Normalised Cut** | **Weighted Kernel $k$-means** |
|---|---|
| $\min \quad \text{Tr}(D^{-1}A) - \text{Tr}(Z^T D^{-\frac{1}{2}} A D^{-\frac{1}{2}} Z)$ | $\min \quad \text{Tr}(WK) - \text{Tr}(Y^T W^{\frac{1}{2}} K W^{\frac{1}{2}} Y)$ |
| $\text{s.t.} \quad \mathcal{X} \in \{0,1\}^{n \times k},$ | $\text{s.t.} \quad \mathcal{X} \in \{0,1\}^{n \times k},$ |
| $\qquad \mathcal{X}1_k = 1_n,$ | $\qquad \mathcal{X}1_k = 1_n,$ |
| $\qquad Z = D^{\frac{1}{2}}\mathcal{X}(\mathcal{X}^T D \mathcal{X})^{-\frac{1}{2}}$ | $\qquad Y = W^{\frac{1}{2}}\mathcal{X}(\mathcal{X}^T W \mathcal{X})^{-\frac{1}{2}}$ |

$$(5)$$

In the above optimisation problems (5), $A$ and $D$ are the adjacency and degree matrices corresponding to some graph $G$, $K$ is a kernel matrix such that $K_{ij} = \langle \phi(x_i), \phi(x_j) \rangle$, and $W$ is a diagonal matrix with $W_{ii} = w(x_i)$. Crucially, these problems are equivalent. Indeed substituting $K = D^{-1}AD^{-1}$ and $W = D$ shows that any normalised cut problem can be written as a kernel $k$-means problem; substituting $D = W$ and $A = WKW$ shows the reverse. The only wrinkle with translating one to the other is that $K$ must be positive semi-definite. If $A$ is not positive semi-definite, then we can add a multiple of $D^{-1}$ to $D^{-1}AD^{-1}$ to make it so while preserving the optimal partitioning (Dhillon et al., 2007).

**Coreset graphs.** The first and third transitions in Figure 1 are due to equivalence (5). The second transition is simply via constructing a coreset, and the fourth is by executing spectral clustering. The crux of our proof is the fifth transition. Specifically, it is not clear how to translate the partition of the coreset (in graph space) to a solution for the entire input graph. The main difficulty with using coresets for the normalised cut problem is that there is no notion of "cluster centers" on graphs. To overcome this, we go back to kernel space (the fifth transition) and label the entire input there. For the rest of the section we prove that this approach preserves the approximation of spectral clustering on the coreset for the entire graph.

To help with this, we will use the following lemma, whose proof is deferred to Appendix B.

**Lemma 5.1** (Adapted from (Kanungo et al., 2002)). *Let $S$ be a finite set of points in a Hilbert space $\mathcal{H}$, and $w : \mathcal{H} \to \mathbb{R}_+$ be the weight of each point. Let $c(S) = \left( \sum_{x \in S} w(x)x \right) / \left( \sum_{x \in S} w(x) \right)$ be the centroid of $S$. Then, it holds for any $z \in \mathcal{H}$ that*

$$\sum_{x \in S} w(x) \|x - z\|^2 = \sum_{x \in S} \left[ w(x) \|x - c(S)\|^2 \right] + |S| \|c(S) - z\|^2 \sum_{x \in S} w(x).$$

Intuitively, the above lemma allows us to draw a connection between the centroids of the coreset partition (in kernel space) and the centroids of the full partition, where every node is assigned to its closest coreset center. We will also make use of the following definitions which allow us to use $k$-partitions instead of set membership matrices in (5).

1. Let $G = (V, E)$ be a graph on $n$ vertices with adjacency matrix $A$ and degree matrix $D$, and let $\Pi$ be a $k$-partition of $V$. Then define $\text{NC}_{A,D}(\Pi) \triangleq \text{Tr}(D^{-1}A) - \text{Tr}(Z^T D^{-1/2} A D^{-1/2} Z)$ to be the objective of the trace minimisation version of the normalised cut problem in (5), where $Z = D^{1/2}\mathcal{X}(\mathcal{X}^T D \mathcal{X})^{-1/2}$ and $\mathcal{X} \in \{0,1\}^{n \times k}$ is the unique set membership matrix on $V$ corresponding to $\Pi$.

2. For any $k$-partition of $X$, $\Pi$, further define $\text{COST}_{K,W}(\Pi) = \text{Tr}(WK) - \text{Tr}(Y^T W^{1/2} K W^{1/2} Y)$ to be the objective of the trace minimisation version of the weighted kernel $k$-means problem in (5), where $Y = W^{1/2}\mathcal{X}(\mathcal{X}^T W \mathcal{X})^{-1/2}$ and $\mathcal{X} \in \{0,1\}^{n \times k}$ is the unique set membership matrix on $X$ corresponding to $\Pi$.

Translating the objective of a $k$-partition with respect to the normalised cut problem to the objective of a set of centres with respect to the kernel $k$-means objective is straightforward. Dhillon et al. (2004) show that for any $k$-partition $\Pi = \{\pi_j\}_{j=1}^k$, we have $\text{COST}_{K_G, W_G}(\Pi) = \text{COST}_w^\phi(V, c_w^\phi(\Pi))$, where $K_G = D^{-1}AD^{-1}$, $W_G = D_G$, $w_G = \text{diag}(W_G)$ and $\phi : V \to \mathcal{H}$ is a map such that

$\langle\phi(x),\phi(y)\rangle = K_G(x,y)$ for all $x,y \in V$. For completeness we include a simpler derivation in Lemma B.1. This implies that $\text{NC}_{A,D}(\Pi) = \text{COST}_{K_G,W_G}(\Pi) = \text{COST}_w^\phi(V, c_{w_G}^\phi(\Pi))$.

The other direction is more difficult; given an arbitrary set of $k$ centers $S = \{s_j\}_{j=1}^k \subset \mathcal{H}$, we would like to construct a $k$-partition $\Pi' = \{\pi_j'\}_{j=1}^k$ of $V$ such that

$$\text{NC}_{A,D}(\Pi') = \text{COST}_{K_G,W_G}(\Pi') = \text{COST}_w^\phi(V, c_{w_G}^\phi(\Pi')) = \text{COST}_w^\phi(V, S).$$

In general it may not be possible to choose $\Pi'$ such that $\text{COST}_w^\phi(V, c_{w_G}^\phi(\Pi')) = \text{COST}_w^\phi(V, S)$. However, the inequality $\text{COST}_w^\phi(V, c_{w_G}^\phi(\Pi')) \le \text{COST}_w^\phi(V, S)$ does hold by choosing $\Pi'$ such that $x \in \pi_j'$ iff $\Delta(\phi(x), S) = \Delta(\phi(x), s_j)$, breaking ties arbitrarily. This is a consequence of Lemma 5.1 which tells us that moving the center of each $\pi'$ to their centroids can only reduce the weighted kernel $k$-means objective. This will be sufficient to prove our main result.

Given a graph $G$ with adjacency matrix $A$ and degree matrix $D$, let $\text{OPTNC}_{A,D}(k)$ denote the optimal value of the normalised cut problem (5) on $G$. We can now state our main result:

**Theorem 1.** *Given a graph $G = (V,E)$ and an $\alpha$-approximation algorithm for the normalised cut problem with $k$ clusters (5), SPECTRALCLUSTERING, Algorithm 3 returns a $k$-partition $\Pi'$ of $V$ such that*

$$\text{NC}_{A_G,D_G}(\Pi') \le \alpha \cdot \frac{1+\varepsilon}{1-\varepsilon} \cdot \text{OPTNC}_{A_G,D_G}(k).$$

*The running time of Algorithm 3 is the sum of the running time of the $\varepsilon$-coreset algorithm, SPECTRALCLUSTERING, and labelling $V$ [2].*

---

**Algorithm 3** CORESET SPECTRAL CLUSTERING

1: **Input:** Graph $G = (V,E)$, $k$ with adjacency matrix $A_G$ and degree matrix $D_G$
2: $K_G, W_G \leftarrow D_G^{-1} A_G D_G^{-1}, D_G$
3: $V', W_H \leftarrow$ An $\varepsilon$-coreset for kernel $k$-means on $(V, K_G, W_G)$
4: $A_H \leftarrow W_H K(V') W_H$           $\triangleright K(V')$ is the principle submatrix of $K$ with respect to $V'$
5: $\Pi \leftarrow$ SPECTRALCLUSTERING$(A_H, k)$                 $\triangleright k$-partition $\{\pi_j\}_{j=1}^k$
6: $\Pi' \leftarrow$ partition assigning each $x \in V'$ to the closest coreset centroid in $c_{w_H}^\phi(\Pi)$.
7: **return** $\Pi'$

---

To prove Theorem 1, we show that the objective of any partition of $V'$ is preserved on $V$ (Lemma B.2) and the objective of any optimal partition of $V$ is preserved on $V'$ (Lemma B.3). We defer their statements and proofs to Appendix B.

*Proof of Theorem 1.* Let $\Sigma = \{\sigma_j\}_{j=1}^k$ be an optimal $k$-partition of $V$ such that $\text{NC}_{A_G,D_G}(\Sigma) = \text{OPTNC}_{A_G,D_G}(k)$ and let $\Sigma' = \{\sigma_j'\}_{j=1}^k$ be the $k$-partition of $V'$ such that $x \in \sigma_j'$ iff $\Delta(\phi(x), c_{w_G}^\phi(\Sigma)) = \Delta(\phi(x), c_{w_G}^\phi(\sigma_j))$, breaking ties arbitrarily . Then we have

$$\text{NC}_{A_G,D_G}(\Pi') \le \frac{1}{1-\varepsilon}\text{NC}_{A_H,D_H}(\Pi) \le \frac{\alpha}{1-\varepsilon}\text{NC}_{A_H,D_H}(\Sigma') \le \alpha\frac{1+\varepsilon}{1-\varepsilon}\text{NC}_{A_G,D_G}(\Sigma),$$

where the first inequality follows from Lemma B.2, the second holds because $\text{NC}_{A_H,D_H}(\Pi) \le \alpha\text{OPTNC}_{A_H,D_H}(k) \le \alpha\text{NC}_{A_H,D_H}(\Sigma')$, and the third follows from Lemma B.3. $\square$

## 6 EXPERIMENTS

We perform three experiments to compare our coreset algorithm against the naive method, and secondly, to compare our Coreset Spectral Clustering algorithm to coreset kernel $k$-means and the sklearn implementation of spectral clustering. Experiments were performed on a dual Intel Xeon E5-2690 system with 384GB of RAM. Our implementation was written in Rust using Faer (El Kazdadi)[3].

---

[2]This can be done efficiently, exploiting sparsity, by first computing the norms of each $c_{w_H}^\phi(\pi_j)$.
[3]A user friendly python wrapper is available here: `github.com/BenJourdan/coreset-sc`.

## 6.1 CORESET RUNNING TIME COMPARISON

We compare the running time of our improved coreset construction with that of Jiang et al. (2024). In particular, we compare the running time of Algorithm 4 using Algorithm 2 (our method) for $D^2$-sampling against Algorithm 4 using Algorithm 1 (Jiang et al., 2024) for $D^2$-sampling. We test these algorithms on the following three large graph datasets from the SuiteSparse Matrix Collection (Davis & Hu, 2011).

- **Friendster**: A snapshot of the social media network with 65M nodes, 1.8B edges, and 5000 labeled overlapping communities.

- **LiveJournal**: A snapshot of the blogging network with 4M nodes, 34M edges, and 5000 labeled overlapping communities.

- **wiki-topcats**: A snapshot of the Wikipedia hyperlink graph with 2M nodes, 28M edges, and 17K overlapping page categories.

We preprocess each graph to be undirected and construct $K = D^{-1}AD^{-1}$ and $W = D$ from the corresponding adjacency and degree matrices. We measure the time it takes each method to construct a 100K point coreset for the weighted kernel $k$-means problem on $K$ and $W$ while varying the number of clusters passed to the $D^2$-sampling routines. Following Jiang et al. (2024), we only do one iteration of importance sampling in Algorithm 4. Figure 2 confirms our method is significantly faster than the previous method. This matches our expectations based on the theoretical results since each graph is very sparse.

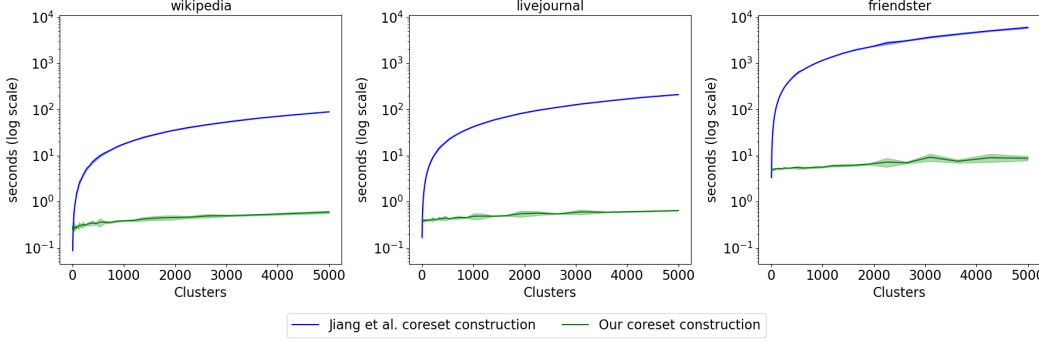

Figure 2: Running time comparison of coreset construction using either Algorithm 1 (Jiang et al., 2024) or Algorithm 2 for $D^2$-sampling. Shaded regions denote 1 standard deviation over 10 runs.

## 6.2 CLUSTERING REAL-WORLD DATASETS

We evaluate the clustering performance of our Coreset Spectral Clustering (CSC) algorithm using our faster coreset construction, as the size of the coreset varies. We test the effect of using both the sklearn spectral clustering algorithm and the faster method of Macgregor (2023) to cluster the coreset graphs. We compare against the sklearn implementation of Spectral Clustering (SC) and coreset kernel $k$-means using the naive coreset construction (Jiang et al., 2024). We measure each algorithm's running time and Adjusted Rand Index (ARI) (Rand, 1971) on nearest neighbour graphs of the following datasets:

- **MNIST**: A labelled collection of 70,000 28x28 pixel images of handwritten digits (Lecun et al., 1998).

- **PenDigits**: 10,992 labeled instances of 2D pen movements covering the digits 0-9, each represented with 16 numerical features. (Anguita et al., 2013).

- **HAR**: A collection of 10,299 labeled smartphone sensor readings capturing six human movements, each represented as 561 numerical features (Anguita et al., 2013).

- **Letter**: A labelled collection of 20,000 instances of handwritten alphabetic english characters, represented as 16 numerical features (Frey & Slate, 1991).

Figure 3 shows the results for the HAR dataset, and demonstrates that the coreset methods run much faster than SC and both variants of CSC perform as well as SC using a coreset less than 5% the size of the original dataset, in a fraction of a second. While increasing coreset size does help coreset kernel $k$-means, it struggles to escape local optima, reducing performance. The results for the other datasets are in Appendix A.

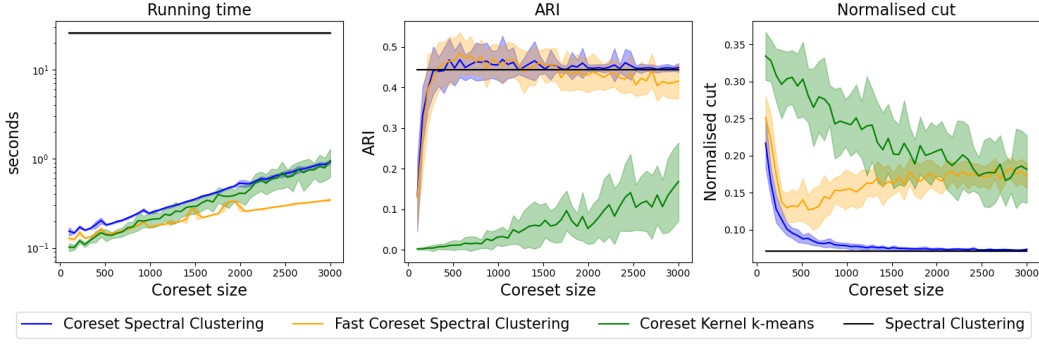

Figure 3: Running time, ARI, and Normalised cut of each algorithm on a 200-nearest neighbour graph of the HAR dataset. Shaded regions denote 1 standard deviation over 20 runs.

### 6.3 CLUSTERING SYNTHETIC GRAPHS WITH MANY CLUSTERS

Finally, we push CSC to the limit by tasking it to cluster the stochastic block model (Abbe, 2018) when the number of clusters is proportional to the number of nodes. We sample graphs where the number of nodes in each cluster is 1000, the probability of an edge between two nodes in the same cluster is $0.5$ and the probability of an edge between two nodes from different clusters is $0.001/k$. We report the ARI of the coreset graph nodes instead of the full graph because the running time is otherwise dominated by labelling. We only consider the variant of CSC using the method of Macgregor (2023) as computing hundreds of eigenvectors becomes prohibitively expensive. The coreset size is set to be $1\%$ the size of the input graph, implying that each cluster has an average of 10 nodes in the coreset graph. Figure 4 shows a gradual decline in performance of fast CSC as $k$ increases, probably due to fluctuations in the number of nodes the coreset samples per cluster. Nevertheless, it still achieves an ARI of 0.5 in less than 3 seconds on a graph with 250K nodes and 250 clusters. On the other hand, local optima render coreset kernel $k$-means useless after 50 clusters.

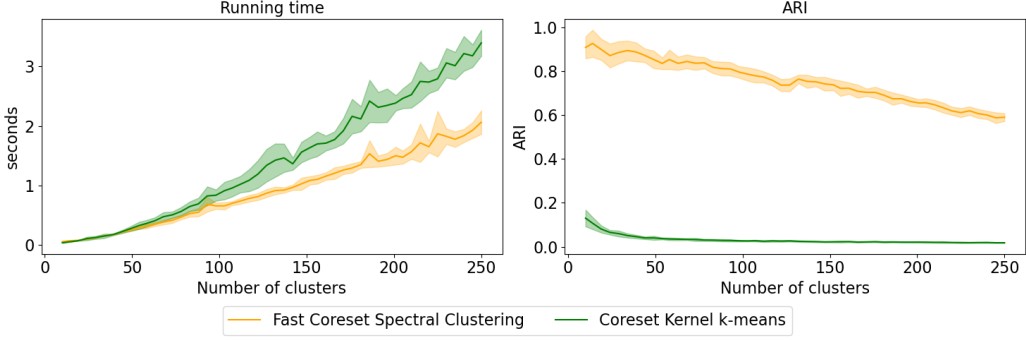

Figure 4: Running time and ARI of each algorithm on the stochastic block model with $k$ clusters of size 1000, $p = 1/2$, $q = 0.001/k$ with a coreset size of $1\%$. Shaded regions denote 1 standard deviation over 20 runs.

## 7 ACKNOWLEDGMENTS

The second author was supported by the following research grants: KAKENHI 21K17703, JST ASPIRE JPMJAP2302 and JST CRONOS Japan Grant Number JPMJCS24K2. The last author was supported by an EPSRC Early Career Fellowship (EP/T00729X/1).

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

# A    FURTHER EXPERIMENTS ON REAL-WORLD DATASETS

In this section, we report the experimental results omitted from Section 6.2.

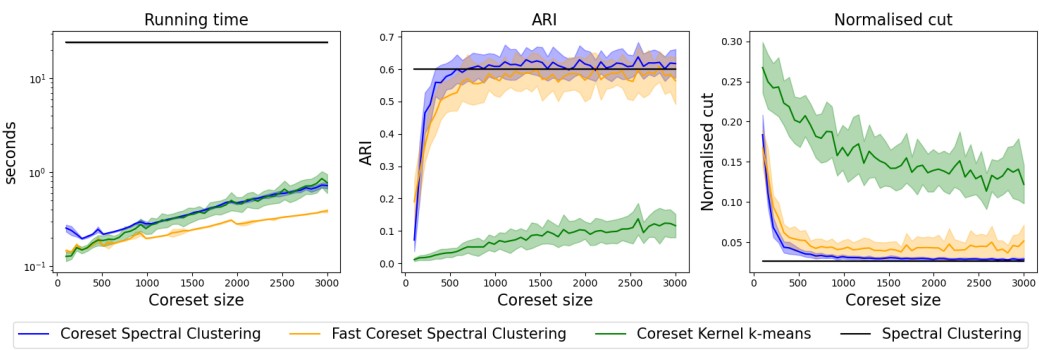

Figure 5: Running time, ARI, and Normalised cut of each algorithm on a 250-nearest neighbour graph of the PenDigits dataset as coreset size varies.

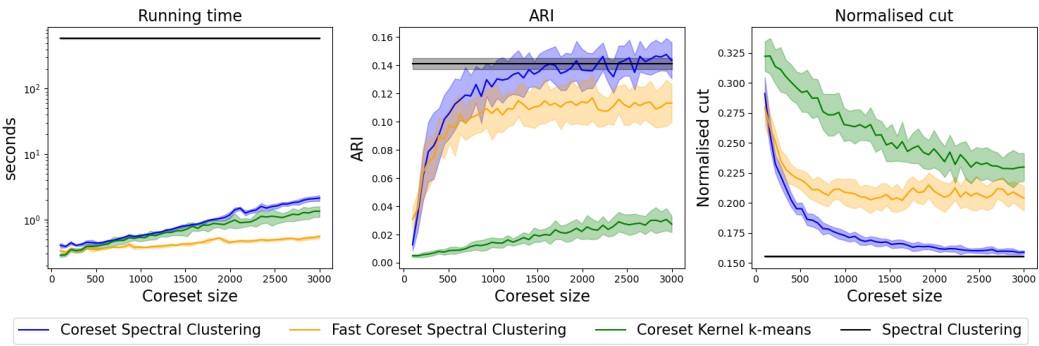

Figure 6: Running time, ARI, and Normalised cut of each algorithm on a 300-nearest neighbour graph of the Letter dataset as coreset size varies.

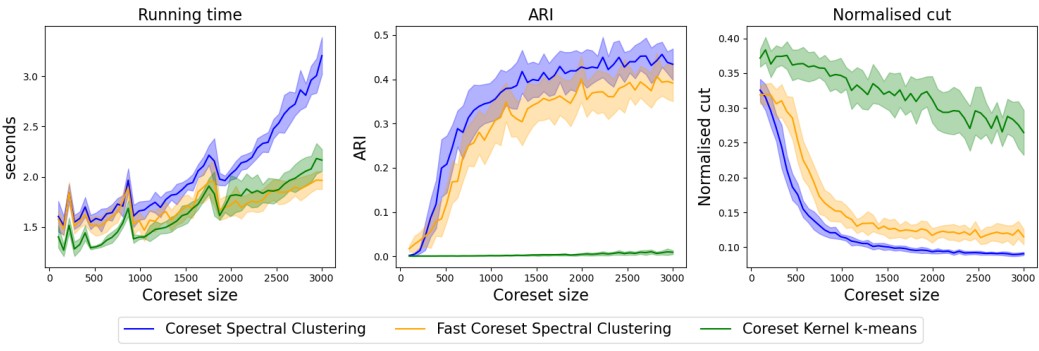

Figure 7: Running time, ARI, and Normalised cut of each algorithm on a 500-nearest neighbour graph of the MNIST dataset as coreset size varies. Sklearn spectral clustering was not included because it took too long.

## B    SUPPLEMENTARY LEMMAS AND PROOFS

In this section we provide the proofs omitted from Section 5.

*Proof of Lemma 5.1.*

$$\sum_{u \in S} w(u) \|u - z\|^2$$

$$= \sum_{u \in S} w(u) \langle u - z, u - z \rangle$$

$$= \sum_{u \in S} w(u) \left\langle \left((u - c(S)) + (c(S) - z)\right), \left((u - c(S)) + (c(S) - z)\right) \right\rangle$$

$$= \sum_{u \in S} w(u) \left( \|u - c(S)\|^2 + 2\langle u - c(S), c(S) - z \rangle + \|c(S) - z\|^2 \right)$$

$$= \sum_{u \in S} w(u) \|u - c(S)\|^2 + 2 \left\langle c(S) - z, \sum_{u \in S} w(u)\left(u - c(S)\right) \right\rangle + |S| \|c(S) - z\|^2 \sum_{x \in S} w(x)$$

$$= \sum_{x \in S} w(x) \|x - c(S)\|^2 + |S| \|c(S) - z\|^2 \sum_{x \in S} w(x)$$

where last step follows from the fact that $c(S)$ is the weighted centroid of $S$, so $\sum_{u \in S} w(u)(u - c(S)) = 0$. $\square$

**Lemma B.1** (Kernel $k$-means objective equivalence). *Given a set of $n$ objects $X$, $w : X \to \mathbb{R}_+$, $K : X \times X \to \mathbb{R}_{\geq 0}$ and $\phi : X \to \mathcal{H}$ satisfying equation 1, for any $k$-partition $\Pi = \{\pi_j\}_{j=1}^k$ of $X$, it holds that*

$$\text{COST}_{K,W}(\Pi) = \text{COST}_w^\phi(X, \{m_j\}_{j=1}^k), \tag{6}$$

*where $m_j = \frac{\sum_{a \in \pi_j} w(a)\phi(a)}{\sum_{b \in \pi_j} w(b)}$ and $W = Diag(w)$.*

*Proof.* Let $X \in \{0, 1\}^{n \times k}$ be the unique set membership matrix corresponding to $\Pi$; that is, $X 1_k = 1_n$ and $X(i, j) = 1$ iff $x_i \in \pi_j$. Therefore $X^T W X = \text{Diag}(s_1, \ldots, s_k)$ where $s_i = \sum_{a \in \pi_i} w(v)$. Expanding the right hand side of equation 6, we have

$$\text{COST}_w^\phi(X, \{m_j\}_{j=1}^k) \triangleq \sum_{j=1}^k \sum_{a \in \pi_j} w(a) \|\phi(a) - m_j\|^2$$

$$= \sum_{j=1}^k \sum_{a \in \pi_j} w(a) \left( \langle \phi(a), \phi(a) \rangle - 2\langle \phi(a), m_j \rangle + \langle m_j, m_j \rangle \right) \tag{7}$$

Notice that $\sum_{j=1}^k \sum_{a \in \pi_j} w(a)\langle \phi(a), \phi(a) \rangle = \sum_{j=1}^k \sum_{a \in \pi_j} w(a) K_{a,a} = \text{Tr}(WK)$ since $\Pi$ is a partition of $X$. Expanding the other terms of equation 7 for the $j$th cluster, we find they are multiples of the same quantity:

$$\sum_{a \in \pi_j} w(a)\langle \phi(a), m_j \rangle = \sum_{a \in \pi_j} \sum_{b \in \pi_j} \frac{w(a)w(b)\langle \phi(a), \phi(b) \rangle}{\sum_{c \in \pi_j} w(c)},$$

and

$$\sum_{a \in \pi_j} w(a)\langle m_j, m_j \rangle = \sum_{a \in \pi_j} w(a) \frac{\sum_{b \in \pi_j} \sum_{c \in \pi_j} w(b)w(c)\langle \phi(b), \phi(c) \rangle}{\left( \sum_{d \in \pi_j} w(d) \right)^2} = \sum_{a \in \pi_j} \sum_{b \in \pi_j} \frac{w(a)w(b)\langle \phi(a), \phi(b) \rangle}{\sum_{c \in \pi_j} w(c)},$$

Finally, we show that these quantities for each cluster are the diagonal entries of the matrix $X^T W K W X (X^T W X)^{-1}$:

$$[X^T W K W X (X^T W X)^{-1}]_{jj} = [X^T W K W X]_{jj} [(X^T W X)^{-1}]_{jj}$$

$$= \left( \sum_a \sum_b X_{aj} X_{aj} [WKW]_{ab} \right) \left( \frac{1}{\sum_{b \in \pi_j} w(b)} \right)$$

$$= \left( \sum_{a \in \pi_j} \sum_{b \in \pi_j} [WKW]_{ab} \right) \left( \frac{1}{\sum_{a \in \pi_j} w(a)} \right)$$

$$= \sum_{a \in \pi_j} \sum_{b \in \pi_j} \frac{w(a)w(b)\langle \phi(a), \phi(b) \rangle}{\sum_{c \in \pi_j} w(c)}.$$

Thus $\text{COST}^\phi_w(X, \{m_j\}) = \text{Tr}(WK) - \text{Tr}(X^TWKWX(X^TWX)^{-1}) = \text{COST}_{K,W}(\Pi)$.

$\square$

**Lemma B.2.** *For any k-partition $\Pi = \{\pi_j\}_{j=1}^k$ of $V'$, we have that*

$$\text{NC}_{A_G, D_G}(\Pi') \leq \frac{1}{1-\varepsilon} \text{NC}_{A_H, D_H}(\Pi),$$

*where $\Pi' = \{\pi_j'\}_{j=1}^k$ is the k-partition of $V$ such that $x \in \pi_j'$ iff $\Delta(\phi(x), c^\phi_{w_H}(\Pi)) = \Delta(\phi(x), c^\phi_{w_H}(\pi_j))$, breaking ties arbitrarily.*

*Proof of Lemma B.2.* We have $\text{NC}_{A_H, D_H}(\Pi) = \text{COST}_{K_H, W_H}(\Pi) = \text{COST}^\phi_{w_H}(V', c^\phi_{w_H}(\Pi))$. Since $V'$ and $w_H$ constitute an $\varepsilon$-coreset, we have that $\text{COST}^\phi_{w_G}(V, c^\phi_{w_H}(\Pi)) \leq \frac{1}{1-\varepsilon}\text{COST}^\phi_{w_H}(V', c^\phi_{w_H}(\Pi))$. Then by the definition of $\Pi'$ and Lemma 5.1, $\text{COST}^\phi_{w_G}(V, c^\phi_{w_G}(\Pi')) \leq \text{COST}^\phi_{w_G}(V, c^\phi_{w_H}(\Pi))$. Since $\text{NC}_{A_G, D_G}(\Pi') = \text{COST}_{K_G, D_G}(\Pi') = \text{COST}^\phi_{w_G}(V, c^\phi_{w_G}(\Pi'))$, the claim follows. $\square$

**Lemma B.3.** *Let $\Sigma = \{\sigma_j\}_{j=1}^k$ be an optimal k-partition of $V$ such that $\text{NC}_{A_G, D_G}(\Sigma) = OPTNC_{A_G, D_G}(k)$. Then*

$$\text{NC}_{A_H, D_H}(\Sigma') \leq (1+\varepsilon)\text{NC}_{A_G, D_G}(\Sigma)$$

*where $\Sigma' = \{\sigma_j'\}_{j=1}^k$ is the k-partition of $V'$ such that $x \in \sigma_j'$ iff $\Delta(\phi(x), c^\phi_{w_G}(\Sigma)) = \Delta(\phi(x), c^\phi_{w_G}(\sigma_j))$, breaking ties arbitrarily.*

*Proof of Lemma B.3.* We have that $\text{NC}_{A_G, D_G}(\Sigma) = \text{COST}^\phi_{w_G}(V, c^\phi_{w_G}(\Sigma))$. Since $V'$ and $w_H$ constitute an $\varepsilon$-coreset, we have that $\text{COST}^\phi_{w_H}(V', c^\phi_{w_G}(\Sigma)) \leq (1+\varepsilon)\text{COST}^\phi_{w_G}(V, c^\phi_{w_G}(\Sigma))$. From the definition of $\Sigma'$ and Lemma 5.1, we have $\text{NC}_{A_H, A_G}(\Sigma') = \text{COST}^\phi_{w_H}(V', c^\phi_{w_H}(\Sigma')) \leq \text{COST}^\phi_{w_H}(V', c^\phi_{w_G}(\Sigma))$. The claim follows. $\square$

## C ALGORITHMS

In this section we provide the coreset kernel $k$-means algorithm given by Jiang et al. (2024) along with our subroutines for Algorithm 2.

### C.1 CORESET KERNEL $k$-MEANS ALGORITHMS (JIANG ET AL., 2024)

---

**Algorithm 4** Constructing an $\varepsilon$-coreset for kernel $k$-means on dataset $X$ with kernel $K$ (Jiang et al., 2024)

---

1: **Input:** $X_0 \leftarrow X, i \leftarrow 0$
2: **repeat**
3:      $i \leftarrow i + 1$ and $\varepsilon_i \leftarrow \varepsilon/(\log^{(i)} \|X_0\|)^{1/4}$             $\triangleright \log^{(i)}(\cdot)$ is the $i$th iterated logarithm.
4:      $X_i \leftarrow \text{IMPORTANCE-SAMPLING}(X_{i-1}, \varepsilon_i)$             $\triangleright$ Algorithm 5
5: **until** $\|X_i\|_0$ does not decrease compared to $\|X_{i-1}\|_0$
6: **return** $X_i$

---

**Algorithm 5** Importance-Sampling($X, \varepsilon$) (Jiang et al., 2024)

---

1: **Input:** $X, \varepsilon$
2: Let $C^* \leftarrow D^2\text{-Sampling}(X)$           $\triangleright$ Algorithm 1 or Algorithm 2 (our faster algorithm)
3: **for** $x \in X$ **do**
4:      $\sigma_x \leftarrow \frac{w_X(x) \cdot \Delta(x, C^*)}{w_X(C)}$
5: **end for**
6: **for** $x \in X$ **do**
7:      $p_x \leftarrow \frac{\sigma_x}{\sum_{y \in X} \sigma_y}$
8: **end for**
9: Draw $N \leftarrow O\left(\frac{k^2 \log^2 k \|X\|_0}{\varepsilon^2}\right)$ i.i.d. samples from $X$, using probabilities $(p_x)_{x \in X}$
10: Let $D$ be the sampled set; for each $x \in D$ let $w_D(x) \leftarrow \frac{w_X(x)}{p_x N}$
11: **return** weighted set $D$

---

## C.2 FAST $D^2$-SAMPLING SUBROUTINES

---

**Algorithm 6** CONSTRUCT$T$

---

1: **Input:** $X$ s.t. $|X| = n, K \in S_{++}^n, W \in \mathbb{R}_+^n$
2: `current_level` $\leftarrow [\,]$
3: **for** $x \in X$ **do**
4:     `leaf`$\leftarrow$ a leaf node corresponding to $x$ with attribute $\Delta \triangleq \langle \phi(x), \phi(x) \rangle + c^*$
5:     `current_level.push(leaf)`
6: **end for**
7: **while** `current_level.len()`$> 1$ **do**
8:     `NextLevel` $\leftarrow [\,]$
9:     **for** $i = 1$ to $2\lfloor \frac{\texttt{current\_level.len()}}{2} \rfloor - 1$ **do**
10:         `left_child` $\leftarrow$ `current_level`$[i]$
11:         `right_child` $\leftarrow$ `current_level`$[i+1]$
12:         `internal` $\leftarrow$ an internal node with `left_child` and `right_child` as their respective children and attribute `contribution` equal to the sum of the contribution of its children.
13:         `next_level.push(internal)`
14:     **end for**
15:     **if** `current_level.len()` is odd **then**
16:         `next_level.push(current_level[current_level.len()])`
17:     **end if**
18:     `current_level` $\leftarrow$ `next_level`
19: **end while**
20: **return `current_level[1]`**

---

**Algorithm 7** REPAIR($x$,$T$)

---

1: Let $L$ be the leaf node corresponding to $x$
2: `delta_difference` $\leftarrow L.\Delta$
3: $L.\Delta \leftarrow 0$
4: **for** Internal node $I$ in the path from $L$ to the root of $T$ **do**
5:     $I.$`contribution` $\leftarrow I.$`contribution` $-$ `delta_difference` $\times w(x)$
6: **end for**
7: **for** $y$ in $N(x)$ **do**
8:     Let $L'$ be the leaf node corresponding to $y$
9:     **if** $\Delta(x,y) < L'.\Delta$ **then**
10:         `delta_difference` $\leftarrow L'.\Delta - \Delta(x,y)$
11:         $L.\Delta \leftarrow \Delta(x,y)$
12:         **for** Internal node $I$ in the path from $L'$ to the root of $T$ **do**
13:             $I.$`contribution` $\leftarrow I.$`contribution` $-$ `delta_difference` $\times w(y)$
14:         **end for**
15:     **end if**
16: **end for**

---

**Algorithm 8** SAMPLEPOINT

---

1: **Input:** $T$
2: Let $I$ be the root of $T$.
3: **while** $I$ is an internal node **do**
4:     Let $C_1$ and $C_2$ be the children of $I$.
5:     `child` $\leftarrow$ Select $C_1$ or $C_2$ with probabilities proportional to $C_1.$`contribution` and $C_2.$`contribution`          $\triangleright$ If $I$ only has one child, sample it with probability 1.
6:     $I \leftarrow$ `child`
7: **end while**
8: **return** the data point associated with $I$.

---

