# OpenReview forum: "Coreset Spectral Clustering"
_ICLR.cc/2025/Conference — ICLR 2025 Poster_

### Official Review · Reviewer_bKLj · 2024-10-31

**Soundness:** 2
**Presentation:** 3
**Contribution:** 2
**Rating:** 3
**Confidence:** 4

**Summary:**

This paper proposes a refined algorithm of constructing a coreset for kernel k-means problem. They improves the time complexity from $\tilde{O}(nk)$ [Jiang et al. ML' 24] to $\tilde{O}(nd_{avg})$, where $d_{avg}$ is the average number of neighbors of a single vertex on the graph defined by the given similarity matrix. They also showed how to use their technique to improve spectral clustering and obtain a approximate solution for normalized cut problem. The experiments are designed to support their theoretical results.

**Strengths:**

The proposed technique of constructing a coreset is quite useful when k is large and the similarity is sufficiently sparse.

**Weaknesses:**

1. Limited contribution. The proposed method highly depends on the former work [Jiang et al. ML' 24]. And their claimed improvements seems trivial. Theorem 1 is also easy to obtain.
2. This paper assumes that the similarity matrix is sparse, which means a vertex has only few neighbors. So when a vertex is sampled, only its neighbors ($d_{avg}$ neighbors on average) need to update the their distance to the sampled set. Therefore, the time complexity of $\tilde{O}(nd_{avg})$ is straightforward.
3. The experimental on the Appendix A seems not ideal. For example, in Figure 5,6,7, the proposed method does not obtain the best ARI; Figure 7 also shows that the green baseline is actually faster. And there is no explanation for that.

**Questions:**

1. In the experimental part, the ARI performance of yours is much better than the green baseline (which is the method of [Jiang et al. ML' 24]). But I think your result is mainly based on the green baseline and you improve their running time. So it makes sense that your method is faster. But why your ARI is so much better than the green baseline?
2. In the experimental part, you mention that you use the nearest neighbor graphs of MNIST. How to construct such a graph on MNIST? Is it a nearest neighbor graphs based on Euclidean distance?

---

> ### Author Response · Authors · 2024-11-15
> **Response**
>
> We thank the reviewer for their time and feedback, and we respond to their concerns below.
>
> On the first question, as we performed our experiments it became clear that coreset kernel clustering struggles with sparse kernels. This was to be expected as Dhillon et al. (2007) reported this phenomenon for kernel $k$-means. They show that the process of ensuring the kernel matrix is positive definite (by adding a multiple of $D^{-1}$ to $K$), makes it more difficult for datapoints to move between clusters. Spectral clustering avoids this phenomenon completely as it recovers the optimal solution of a relaxed problem which is invariant under this shift.
>
> For the second question, yes. We construct a similarity graph based on the Euclidean distance between the input instances.
>
> Regarding Figures 5, 6, and 7, we experimentally evaluate two versions of CSC: the ordinary CSC algorithm, and a fast version which uses the spectral clustering variant proposed by Macgregor (2023). We find that both CSC algorithms perform better than the coreset kernel $k$-means baseline in terms of ARI and at least one of our variants meets the ARI of standard spectral clustering with a suitably large coreset. The fast CSC variant offers a significant speedup at the cost of slightly lower ARI. This speedup becomes more visible as the number of clusters increases.
>
> On your point about Figure 7, if the partition of coreset kernel $k$-means doesn't change from one iteration to the next (as was often the case) then the algorithm terminates early. Based on the work of Dhillon et al. (2007), this is to be expected for sparse graphs. Note that the baseline performs poorly in terms of ARI in this instance.

---

> > ### Comment · Reviewer_bKLj · 2024-11-19
> > **Feedback**
> >
> > I thank the authors for addressing my questions on the experiments. I also read the comments from other reviewers. My concerns on the novelty (W1 and W2) are also mentioned by another reviewer, but unfortunately the authors did not directly explain that in the rebuttal. I agree that this is not a bad paper, and provides some interesting understanding on spectral clustering. But given the novelty concerns and the top bar of ICLR, as one of the top three ML conferences, I decide to keep my previous judgement.
> >
> > Btw, I notice that there is a reviewer who gives an extremely high grade "strong accept". In my personal opinion, speak frankly, no matter whether this paper should be accepted or not, the current result is far away from a strong accept paper for ICLR.

---

> > > ### Comment · Reviewer_8Sqn · 2024-11-20
> > >
> > > I read the response of bKLj which included a criticism of my review.
> > > From bKLj review it is clear that they completely miss the contribution of the
> > > paper, the interplay between spectral clustering and kernel kmeans.
> > >
> > > Dhillon04, further elaborated in Dhillon07, proved the equivalence of kernel
> > > kmeans and spectral clustering.  It was expected at that time that this would
> > > lead to many new graph clustering algorithms. This didn't happen because kmeans
> > > gets stuck in a local minimum.
> > >
> > > This paper shows an interesting approach to avoiding this problem, and I view it
> > > as giving a solution to an important problem that was open for 20 years.
> > >
> > > My point is that  bKLj didn't understand the paper, and apparently didn't
> > > understand the authors rebuttal. As such, their ranking is irrelevant.

---

> > > > ### Comment · Reviewer_bKLj · 2024-11-20
> > > > **Further feedback**
> > > >
> > > > Dear 8Sqn,
> > > >
> > > > Firstly, I would like to express my apologies if my previous comment lets you feel uncomfortable. I don't want to criticize  anyone. I only want to present my own opinion regarding this paper.
> > > >
> > > > Now, let my focus on evaluating this paper from theory and experiments separately.
> > > >
> > > > **For theory:** The main contribution of this paper, Theorem 1, is quite easy to obtain. If you read the proof (also the proofs of B.2 and B.3), you will see this is a straightforward result from the definition of coreset (definition 4) and Lemma 5.1 (this is Kanungo et al., 2002; actually this is a very basic property of k-means, and widely used in many k-means papers). In my opinion, the main value of this paper lies in Sec 4.1, which is a slight modification of [Jiang et al 2024]'s coreset method on sparse graph. So, at least in theory, I cannot agree that this is an important result.
> > > >
> > > > **For experiment:** Yes, I agree that the paper exhibits good empirical performance on clustering, though it only considers two evaluation metrics ARI and normalized cut.
> > > >
> > > > Overall, the above is my judgement on this paper, and I am open to any further discussion. Have a nice day!
> > > >
> > > > bKLj

---

### Official Review · Reviewer_in36 · 2024-11-03

**Soundness:** 3
**Presentation:** 3
**Contribution:** 2
**Rating:** 6
**Confidence:** 2

**Summary:**

The paper tackles the challenges of clustering large, sparse datasets, where traditional spectral clustering methods can be computationally demanding. While spectral clustering is widely used for identifying non-linear cluster boundaries, its dependence on dense similarity matrices restricts scalability, particularly when dealing with numerous clusters. The authors introduce Coreset Spectral Clustering (CSC), a method that merges the efficiency of coreset sampling with the accuracy of spectral clustering, achieving a substantial speedup while maintaining clustering precision.

**Strengths:**

- CSC is optimized for sparse graphs, where the sparsity structure significantly reduces both computation and memory usage. By using a small, representative subset of data (the coreset), CSC scales well with data size and can handle graphs with millions of nodes and thousands of clusters. This scalability makes CSC suitable for large datasets in social networks, biological clustering, and sensor network analysis, where traditional methods would struggle.

- Standard spectral clustering can become infeasible with large, dense similarity matrices due to the high demands on computation and memory. CSC addresses this by working with a sparse kernel matrix and clustering only on the coreset, significantly reducing matrix size and computational cost. This efficiency enables CSC to process large datasets on standard hardware, which would otherwise require extensive resources for traditional spectral clustering.

- A smaller coreset speeds up computation, while a larger coreset captures more nuances in the data structure. This adaptability is useful for applications with specific accuracy or runtime needs, making CSC versatile across different types of data and clustering goals.

**Weaknesses:**

- The accuracy of CSC’s clustering largely depends on the representativeness of the coreset. To achieve high-quality clusters, the coreset need to accurately capture key structural and distributional aspects of the dataset. In datasets with uneven distributions or subtle data patterns, it could be difficult to create a coreset that fully represents the original data, and even minor inaccuracies could impact clustering results.

- CSC relies on an initial similarity or nearest-neighbor graph, and parameters such as the number of neighbors (k) or distance threshold (ϵ) can significantly affect clustering performance. Choosing suboptimal values for these parameters may lead to an inaccurate initial graph structure, impacting the quality of the final clusters.

**Questions:**

See weakness.

---

> ### Author Response · Authors · 2024-11-15
> **Response**
>
> We thank the reviewer for their time and feedback, and we respond to their concerns below.
>
>
> For the first question, our results hold regardless of the distribution of datapoints. This is due to the nature of the underlying coreset guarantee for kernel $k$-means: for any dataset, a coreset preserves the objective of every set of centers with high probability. Most coreset algorithms achieve this by performing some sort of importance sampling to make sure they cover imbalanced or uneven clusters sufficiently.
>
>
> To answer the second question, we agree that the performance of a clustering algorithm depends on the chosen parameter when constructing a similarity graph and, rather than a downside of our technique, most clustering algorithms would face this scenario. However, there have been many empirical studies on choosing the right parameters for typical datasets, and we notice that the value of $k$ between $200$ and $500$ as the number of neighbours suffices for our experiments.

---

> > ### Comment · Reviewer_in36 · 2024-11-26
> >
> > I thank the authors for the responses. I shall keep my score unchanged.

---

### Official Review · Reviewer_CV9w · 2024-11-04

**Soundness:** 4
**Presentation:** 4
**Contribution:** 3
**Rating:** 8
**Confidence:** 3

**Summary:**

This paper presents an algorithm coreset spectral clustering algorithm for k-means clustering. This is done by first converting the input graph into a k-means problem instance, constructing and $\epsilon$-coreset for this instance, then solving the spectral clustering problem on the reduced graph. A second contribution is an algorithm for fast $D^2$-sampling utilized in coreset construction, which results in an coreset construction algorithm with running time $\widetilde{O}(n d_{avg})$.

**Strengths:**

- The contribution of the paper is solid, with the main idea being combining the approaches of coreset construction and spectral clustering. The utilization of sparsity to improve the running time of clustering algorithm is also well-executed.
- The presentation is overall excellent, with all of the contributions stated clearly. Schemes and easy-to-read pseudocode are very helpful with understanding the approach.
- The experimental section is detailed and well-organised.

**Weaknesses:**

It is somewhat unclear how often it is desired to solve spectral clustering on sparse data, or whether settings of interest have $d_{avg} < k$. I would like the authors to add an overview on how clustering methods are used in the empirical research, for example social network analysis, in the introduction or related work.

**Questions:**

Please address the concern that I raised in the weaknesses section.

---

> ### Author Response · Authors · 2024-11-15
> **Response**
>
> We thank the reviewer for their time and positive review, and we respond to their comments below.
>
> Regarding the comments on $d_{avg}$ vs $k$, we notice that most graphs occurring in practice, including internet graphs, biological networks, and several social networks, are power-law graphs with very low average degrees. However, the number of clusters of these graphs could be quite high as shown in the following examples from our first experiment:
>
> - The Friendster graph has an average degree of 28 while the number of ground truth (overlapping) communities is 5,000.
> - The LiveJournal graph has an average degree of 9 and 5,000 ground truth communities.
> - The wiki-topcats graph has an average degree of 14 and 17,000 ground truth communities.
>
> Hence, for real-world graphs, it is typical that the number of clusters dominates the average degree. The actual running time of our algorithm is $\widetilde{O}(n\cdot \min (k, d_{avg}))$, which is always asymptotically better than the method of Jiang et al. We will make it clearer in the abstract of the next version of the paper.
>
> Finally, we agree that adding an overview of clustering methods in empirical research would
> enrich our discussion, and we couldn't do so due to the page limit. We will add more such discussions in the related work section in the next version of our paper.

---

### Official Review · Reviewer_8Sqn · 2024-11-05

**Soundness:** 4
**Presentation:** 4
**Contribution:** 3
**Rating:** 10
**Confidence:** 4

**Summary:**

The paper leverages the equivalence between kernel kmeans and spectral
clustering to improve spectral clustering. As a secondary result they also
improve coreset construction for sparse matrices.

The equivalence between kernel kmeans and spectral clustering is well known.
It is, therefore, natural to expect improvements in kernel kmeans
algorithms to produce better spectral clustering. Results along this line
were recently described by Jiang24, performing kernel kmeans on
weighted sampled data (coreset).

The paper argues that improving coresets do not necessarily lead to
improved spectral clustering because the kernel kmeans typically gets
stuck in a local minimum. By contrast, spectral clustering computes
an approximation to the global optimum, and does not gets stuck in local
minima.

Using this key observation the authors propose a novel framework of going
back and forth between
the graph and the points in high dimensional space that are represented
by the coreset. This improves the speed, but not the quality of the clustering
(as measured by NCUT). The paper shows that the reduction in quality
is linear.

**Strengths:**

The paper is very nicely written. It describes a result that appear interesting
in theory and useful in practice.

**Weaknesses:**

An important side result is the fast construction of a coreset that can be used
for kernel k-means clustering. The improvement comes from a fast
$D^2$ sampling technique. I believe that there are other, competitive
fast sampling techniques and I was missing a comparison.
Here is an example:

Chib and Greenberg, 1995, Understanding the Metropolis-Hastings algorithm,
The American Statistician.


In addition please see the questions below.

**Questions:**

The result: Why are the derivation and experiments discussing only NCUT?
The equivalence of Dhillon04 was extended in Dhillon07 to other criteria,
in particular RatioCut. It should also apply to the newer stochastic box model.


Experimental results: why is there no comparison of the NCUT values
that were obtained in the experiments? The current evaluation is
in terms of ARI, but this is not what the algorithms attempt to maximize.

---

> ### Author Response · Authors · 2024-11-15
> **Response**
>
> We thank the reviewer for their time and insightful comments, and we respond to their mentioned points below.
>
> First of all, we think it is a very interesting idea to apply other sampling techniques, including the Metropolis-Hasting algorithm, to improve our result. However, applying these algorithms would require us to significantly expand our analysis and proof. We will leave this for further work.
>
> For your comments on the RatioCut, we believe that a similar result should hold for the RatioCut as well, and we chose to use NCut since it's arguably more commonly used in spectral clustering literature. We can report our result with respect to RatioCut in the final version if the reviewer thinks it is necessary.
>
> Regarding your last question, ARI is commonly used to compare the performance of clustering algorithms that optimise different objectives as long as ground-truth labels are available. For label-free objectives such as NCUT and RatioCut, there may not even be a uniquely defined optimal clustering. Even though the algorithms we test do not optimise ARI directly, ARI allows us to evaluate them fairly.

---

> > ### Comment · Reviewer_8Sqn · 2024-11-17
> >
> > I would like to thank the authors for their paper which I consider
> > a major contribution.
> >
> > In responding to my review the authors say:
> >
> > "We can report our result with respect to RatioCut in the final version if the
> > reviewer thinks it is necessary."
> >
> > No, I do not. I just wanted to make the point that I believe
> > the result extends to
> > all clustering criteria proved in Dhillon07 to be equivalent to kernel k-means.
> >
> > On the other hand I do not consider the authors response to my query about
> > the lack of experiments with ncut to be satisfactory. Let me explain:
> >
> > The strength of the result is in its theoretical guarantees on ncut
> > minimization, related to both time and accuracy. The critical question that I
> > (and others may) have is: Does this lead to a better practical ncut
> > minimization, or is it just the result of a careful analysis of an inferior
> > algorithm?
> >
> > This can be easily resolved in experiments. But the paper reports NO experiments
> > with ncut. We are expected to somehow believe that ARI accurately
> > reflects ncut. But this is very questionable.
> > ARI relates to supervised learning, and ncut to unsupervised learning.
> > See, e.g., the paper by Farber, "On using class-labels in evaluation of
> > clusterings", 2010.
> >
> > To push the point further, the authors argue that ARI allows them to "evaluate
> > their results fairly." I don't think this is the case. If we take the algorithm
> > by itself without its theoretical guarantees on ncut, then it requires
> > significantly more experimental work to argue that it compares favorably with
> > all other graph clustering algorithms. On the other hand, demonstrating the
> > results on ncut to validate the theoretical results can be easily achieved.
> >
> > I wish to point out that I will keep my high score regardless of the authors
> > response.

---

> > > ### Author Response · Authors · 2024-11-18
> > > **Response**
> > >
> > > Thank you for reading our response. Following your last comment, we added
> > >  the normalised cut metric comparison for real-world datasets in the second experiment. These results are reported in Figures 3(c), 5(c), 6(c), and 7(c) of our updated submission.
> > >
> > >  Our experiments show that, under the NCut metric, the CSC variant with standard spectral clustering (SC) as subroutine consistently outperforms coreset kernel $k$-means, and the performance of the CSC variant with fast spectral clustering is between the CSC with standard SC and coreset kernel $k$-means. We remark that our experiment applied the default implementation of fast SC by Macgregor (2003), and we believe that with different parameters our algorithm with fast SC could result in better performance for the tested datasets.

---

### Official Review · Reviewer_Gaz4 · 2024-11-06

**Soundness:** 3
**Presentation:** 3
**Contribution:** 3
**Rating:** 6
**Confidence:** 3

**Summary:**

The paper develops new tools in coreset construction merging ideas from two different problems: one is the coresets for k-means and kernel k-means clsutering and the other is spectral clustering, hence the name Coreset Spectral Clustering of the paper.

The main result is to give an approximation algorithm for the problem of normalized cut based on coresets. Specifically, they can approximately solve the problem on the coreset graph and prove that this is enough to get a reasonable approximation on the original input graph. The authors also perform experiments and demonstrate that their approach leads to asymptotically faster results on large real-world graphs with many clusters beating prior coreset kernel k-means approaches for sparse kernels.

The second result of the paper is to speed up the running time of the current state-of-the-art coreset algorithm for the problem of kernel k-means on sparse kernels, where the speed up depends on the average degree of the graph.

**Strengths:**

-nice idea to rely on kernel sparsity that yields the first coreset construction for kernel spaces and leads to speed up which is especially useful for large graphs with many clusters.

-the two main protagonists here which are spectral clustering and kernel k-means are studied often separately, and I view this approach of merging ideas/techniques interesting.

-the coreset spectral clustering algorithm is interesting and gives a clean result statement: an \alpha-approximation of the normalized cut problem on the coreset graph will  in fact give an O(\alpha)-approximation of the normalized cut problem on the original graph. To me this is a very useful and interesting statement as it can be used as a black box and lead to practical results as well.

**Weaknesses:**

-novely: while the paper draws inspiration and combines cleverly prior works on normalized cut, kernel clustering and coresets, I wanted to point out that the current paper seems to heavily rely on ideas and techniques that were developed before. Of course the authors had to cleverly combine them in order to get the clean statement as their main result. I also read parts of the technical proofs in the appendix, and I believe that in terms of techniques the paper is a bit weak. Perhaps the authors could elaborate on what crucial ideas in terms of techniques were the novel aspects of this work. Specifically the analysis of Jiang et al. seems to be doing the heavy lifting in many parts of the paper, and conditioned on that  paper, I believe  the current technical contribution appears to be slightly less solid. This is my only concern about the paper, otherwise I do like the paper.

**Questions:**

-The authors say their speed up is from nk to nd where d is the average degree in some sense. In the abstract this is a bit confusing: but is this necessarily a speedup; what if k is relatively small but the average degree in the kernel matrix leads to more non-zero entries? Perhaps this is good to clarify early on as you do later in the main body cause the reader might be confused.

-While reading the paper, many ideas used in the analysis are actually coming from (and are cited) from the prior work by Jiang et al. I was curious if the authors could elaborate on what ideas were the novel part of the paper?

---

> ### Author Response · Authors · 2024-11-15
> **Response**
>
> We thank the reviewer for their time and feedback, and we respond to points below.
>
> On the first question, the actual running time of our algorithm is $\widetilde{O}(n\cdot \min (k, d_{avg}))$, which is always asymptotically better than the method of Jiang et al. We will make it clearer in the abstract of the next version of the paper.
>
> On the second question, we highlight that our technical contribution is to develop the relationship between the approximation guarantee from cluster centers in (coreset) kernel space and the one for graph partition.  To achieve this, we employ the techniques from kernel $k$-means and spectral clustering, the two fields which had been studied separately in the past.

---

> > ### Comment · Reviewer_Gaz4 · 2024-11-17
> > **response to rebuttal**
> >
> > I have read your response and I will keep my score.

---

### Meta-Review · Area_Chair_Agn8 · 2024-12-20

**Metareview:**

This paper presents several interesting results related to coresets for clustering, and specifically spectral clustering. The authors have one algorithmic result that improves on the coreset construction speed of prior work by Jiang et al. in the important setting where the similarity matrix is sparse. While this result does not require a huge amount of work over the prior result, the new algorithm is interesting and non-obvious. Second, the authors present a result that shows that corests for kernel k-means can be used *directly* within a spectral clustering algorithm to speed up the method. Again, reviewers found this result interesting and new, even if the proof is relatively straightforward. On balance, this paper is well-written, and provides new results that should be of interest to the ICLR community.

**Additional Comments On Reviewer Discussion:**

The discussion phase was helpful. With a score of 10, one reviewer is clearly off the mark, and this was pointed out by another reviewer who provided more balanced feedback. I discounted the score of 10 in making my decision.

---

### Decision · Program_Chairs · 2025-01-22

Accept (Poster)